# Response of Breast Cancer Cells to PARP Inhibitors Is Independent of BRCA Status

**DOI:** 10.3390/jcm9040940

**Published:** 2020-03-30

**Authors:** Man Yee Keung, Yanyuan Wu, Francesca Badar, Jaydutt V. Vadgama

**Affiliations:** 1Division of Cancer Research and Training, Charles R. Drew University of Medicine and Science, Los Angeles, CA 90059, USA; manyeekeung@cdrewu.edu (M.Y.K.); badarf@bu.edu (F.B.); 2David Geffen UCLA School of Medicine, Jonsson Comprehensive Cancer Center, University of California at Los Angeles, Los Angeles, CA 90095, USA

**Keywords:** PARP inhibitors, Triple-negative breast cancer, BRCA1/2 mutations, HER2-positive breast cancer

## Abstract

Poly (ADP-ribose) polymerase inhibitors (PARPi) have proven to be beneficial to patients with metastatic breast cancer with BRCA1/2 (BReast CAncer type 1 and type 2 genes) mutations. However, certain PARPi in pre-clinical studies have been shown to inhibit cell growth and promote the death of breast cancer cells lacking mutations in BRCA1/2. Here, we examined the inhibitory potency of 13 different PARPi in 12 breast cancer cell lines with and without BRCA-mutations using cell viability assays. The results showed that 5 of the 8 triple-negative breast cancer (TNBC) cell lines were susceptible to PARPi regardless of the BRCA-status. The estrogen receptor (ER) negative/ human epidermal growth factor receptor 2 (HER2) positive (ER-/HER2+) cells, SKBR3 and JIMT1, showed high sensitivity to Talazoparib. Especially JIMT1, which is known to be resistant to trastuzumab, was responsive to Talazoparib at 0.002 µM. Niraparib, Olaparib, and Rucaparib also demonstrated effective inhibitory potency in both advanced TNBC and ER-/HER2+ cells with and without BRCA-mutations. In contrast, a BRCA-mutant TNBC line, HCC1937, was less sensitive to Talazoparib, Niraparib, Rucaparib, and not responsive to Olaparib. Other PARPi such as UPF1069, NU1025, AZD2461, and PJ34HCl also showed potent inhibitory activity in specific breast cancer cells. Our data suggest that the benefit of PARPi therapy in breast cancer is beyond the BRCA-mutations, and equally effective on metastatic TNBC and ER-/HER2+ breast cancers.

## 1. Introduction

Breast cancer is one of the most common cancers worldwide, and the leading cause of death in women [1]. Therapeutic failure and distant metastasis have been significant challenges in the treatment of breast cancer as well as the leading cause of mortality in breast cancer patients. Compared to all different types of breast cancers, the treatment of triple-negative breast cancer (TNBC) remains challenging due to the disease’s aggressiveness and limited target therapies [2]. Among various ethnic groups, African Americans, especially younger African Americans, are more likely to have TNBC that contributes significantly to increased mortality and cancer health disparities [3,4]. Another aggressive type of breast cancer is human epidermal growth factor receptor 2 positive (HER2+) breast cancer. Until the discovery and application of trastuzumab for the treatment for HER2+ breast cancer, patients with HER2+ tumors had inferior disease outcomes [5,6]. However, almost 52% of HER2+ patients will fail trastuzumab treatment, leading to disease progression [7].

Poly (ADP-ribose) polymerase inhibitors (PARPi) are targeted therapies that inhibit PARP proteins, which are involved in the repair of single-strand DNA. Recently, several PARPi have been approved by the FDA (Food and Drug Administration) to treat different cancers, which include metastatic TNBC and estrogen receptor negative (ER-)/HER2+ breast cancer with BRCA (BReast CAncer type 1 and type 2 genes) -mutations.

The BRCA1 gene on human chromosome 17q21 has multiple functions in DNA repair, including recognition of DNA damage, checkpoint activation and recruitment of DNA repair protein in cell growth, cell division, and the repair of damage to DNA [8,9]. The BRCA2 gene on chromosome 13 has a function in the recruitment of RAD51, DNA Repair Protein RAD51 Homolog 1, to double-stranded DNA breaks to allow the homologous recombination (HR) repair [9,10]. Mutations in BRCA genes are responsible for most cases of early-onset hereditary breast and ovarian cancers [11]. The BRCA1 mutations account for approximately 5% to 10% of all breast cancer [12]. Around 300 mutations in the BRCA1 gene have been identified that included insertions, deletions, and nonsense mutations, most of them lead to functionally inactive proteins [8,13]. The region of mutations could vary among ethnic/race groups, such as Caucasians and African Americans [14]. Breast cancer patients with BRCA1 mutations are more frequently found to have TNBC [15]. In a typical setting, the BRCA genes are key players in the homologous recombination pathway, the principle double-strand break repair mechanism [16]. In the event of a BRCA mutation, DNA damage can be repaired by alternate mechanisms such as the PARP pathway, thereby maintaining cell viability. Therefore, when PARP is inhibited in a BRCA-deficient setting, DNA damage accumulates, and cytotoxicity results. This synthetic lethality is mediated by the absence of high-fidelity repair mechanisms, making PARPi a promising therapeutic approach for BRCA-mutant tumors [17]. 

The monotherapy of PARPi has demonstrated promising results in specific patient cohorts [18,19,20]. However, clinical evidence has shown that mutations in BRCA do not entirely account for the treatment benefit from PARPi. Recent data from a phase III trial has demonstrated the interest of PARPi to recurrent ovarian cancer regardless of the presence or absence of BRCA mutations or homologous recombination deficiency (HRD) [13,14]. Pre-clinical studies have also revealed the potential of PARPi in promoting the death of breast cancer cells lacking mutations in BRCA1 or BRCA2 [21,22]. 

Decreased expression of the BRCA1 gene and protein has also been seen in sporadic breast cancer. Loss of BRCA1 nuclear protein was found in a large proportion of sporadic breast cancer tissues, and loss of both cytoplasmic and nuclear BRCA1 protein was observed in approximately 19% of sporadic breast cancer tissues [23,24]. Except for methylation of the BRCA promoter, sporadic loss of the wild-type allele is the standard mechanism of BRCA inactivation [23]. The loss of heterozygosity of BRCA1 in somatic breast cancer has been shown to share genotype/phenotype features with familial breast cancer and have a defect of the DNA repair pathway [24,25]. BRCA1 allelic loss also has been found in the most breast cancer cells with or without mutations, and the allelic loss affected BRCA1 transcript expression [26]. Therefore, we hypothesize that PARPi could also effectively treat breast cancer without BRCA mutations.

Thus, the field of PARPi is rapidly expanding, and generations of PARPi have been developed. The various PARPi have a different chemical structure that leads to differences in pre-clinical and clinical potency. Hence, a study with appropriate design to evaluate the distinct PARPi’s pre-clinical influence in cancer cells with different BRCA status and their potential doses is needed. Our research is designed to assess the efficacy of different PARPi in the treatment of different subtypes of breast cancer, which include cells that have mutant or wild-type BRCA. We are also examining the effect of PARPi on cancer cells from advanced and early-stage breast cancers with the different status of ER/PR (progesterone receptor) and HER2 receptors. Our data confirm that the sensitivity to PARPi is not only dependent on BRCA status.

Furthermore, the inhibitory potency of PARPi is highly promising for the treatment of HER2+ tumors and for HER2+ tumors which become resistant to trastuzumab. The study also revealed the commonalities and differences in sensitivity to various PARPi to breast cancer with different ER/PR/HER2 and BRCA mutation status. In summary, our data suggest that PARPi therapy may be beneficial to a broader range of breast cancers. Our study provides pre-clinical evidence toward the future development and selection of PARPi for treating breast cancer patients.

## 2. Materials and Methods

### 2.1. Reagents

The PARP inhibitors used in this study were purchased from Selleckchem (Houston, TX, USA) with the following names and catalog numbers: A-966492 (catalog number S2197), AG14361 (catalog number S2178), AZD2461 (catalog number S7029), E7449 (catalog number S8419), G007-LK (catalog number S7239), Niraparib (MK-4827; catalog number S2741), NMS-P118 (catalog number S8363), NU1025 (catalog number S7730), Olaparib (AZD2281, Ku-0059436; catalog number S1060), PJ34-HCl (catalog number S7300), Rucaparib (AG-014699, PF-01367338; catalog number S1098), Talazoparib (BMN673; catalog number S7048), and UPF1069 (catalog number S8038). Stock solutions were dissolved in dimethyl sulfoxide (DMSO) according to the manufacturer’s protocol to a concentration of 10 mM and stored at −20 °C. Table 1 lists the PARP inhibitors used in this study and their proposed targets.

### 2.2. Cell Lines and Cell Culture

Cell lines were purchased from American Type Culture Collection (ATCC: Manassas, VA, USA). Culture media was purchased from Gibco (ThermoFisher Scientific: Waltham, MA, USA) and VWR Life Science (Avantor: Radnor, PA, USA). BT474 (Luminal B; ER+, PR+, HER2+), MCF7 (Luminal A; ER+, PR+, HER2-), MDA-MB-231 (mesenchymal-like; ER-, PR-, HER2-), MDA-MB-436 (BRCA1-mutant, Mesenchymal-like; ER-, PR-, HER2-), MDA-MB-468 (Basal-like; ER-, PR-, HER2-), and SKBR3 (ER-, PR-, HER2+) were cultured in Dulbecco’s modified Eagle’s medium (DMEM)/F12 supplemented with 10% fetal bovine serum (FBS), 2 mM L-glutamine, and 100 U penicillin/0.1 mg/mL streptomycin. BT549 (Mesenchymal; ER-, PR-, HER2-), HCC1143 (Basal-like 1; ER-, PR-, HER2-), HCC70 (Basal-like; ER-, PR-, HER2-), HCC1806 (Basal-like; ER-, PR-, HER2-), and HCC1937 (BRCA1-mutant, Basal-like 1; ER-, PR-, HER2-) were cultured in RPMI-1640 media supplemented with 10% FBS, 2 mM L-glutamine, and 100 U penicillin/0.1 mg/mL streptomycin. JIMT1 (trastuzumab-resistant, ER-, PR-, HER2+ breast cancer cell line) was obtained from DMSZ (German Collection of Microorganisms and Cell Cultures). It was cultured and maintained in Dulbecco’s modified Eagle’s medium (DMEM)/F12 supplemented with 10% FBS, 2 mM L-glutamine, and 100 U penicillin/0.1 mg/mL streptomycin. All cell lines were grown and maintained at 37 °C with 5% CO_2_. Cell cultures were passaged routinely, detached using 0.25% trypsin-EDTA solution, and counted using Cellometer Vision CBA (Nexcelom: Lawrence, MA, USA). Logarithmically growing cells were used in all experiments. 

### 2.3. Cell Viability Assays

Exponentially growing cells were seeded at 20,000–60,000 cells/mL in 96-well plates and incubated overnight to facilitate cell attachment. The following day, the cell cultures were treated with PARP inhibitors in concentrations ranging from 0.001 to 200 µM. Cell growth was determined by the addition of 3-(4,5-dimethylthiazol-2-yl)-2,5-diphenyl tetrazolium bromide (MTT; Promega: Madison, WI, USA) 7 days after continuous exposure to the drug. 50 µl of 1 mg/mL MTT reagent was added to each well and incubated for 4 h in a CO_2_-free 37 °C incubator. Following incubation, MTT reagent was removed from each well and replaced by 100 µl of DMSO. The conversion of MTT to purple formazan by viable cells was measured at an absorbance of 560 nm using the Glomax Microplate Reader (Promega: Madison, WI, USA). Growth curves represent a percent of cell growth after treatment with various concentrations of PARPi treatment. Assays with 12 different cell lines and 13 different PARP inhibitors were repeated at least three times. 

### 2.4. Statistical Analysis and the Half Maximal Inhibitory Concentration (IC_50_) Calculations

First, the mean values of cell viability at each treatment dose from 3 independent assays were performed, and then the values were logarithm-transformed. The final value of IC_50_ was determined by using the semi logarithm-transformed dose-response data with a Linear Regression model. Statistical analysis was performed using SPSS Statistics software (IBM; Armonk, NY. USA). 

## 3. Results

The PARPi used were selected for specificity against PARP1, PARP2, PARP3, PARP5a/b, or a combination of these (Table 1). The in vitro effect of 13 PARPi: A-966492, AG14361, AZD2461, E7449, G007-LK, Niraparib, NU1025, NMS-P118, Olaparib, PJ34-HCl, Rucaparib, Talazoparib, and UPF 1069 was tested on the selected 12 established human breast cancer cell lines (Table 2). Among those PARPis, Niraparib, Olaparib, Rucaparib, and Talazoparib have been approved by the FDA for treating advanced ovarian, pancreatic, and breast cancers [34]. Olaparib and Talazoparib have been approved as monotherapy in HER2-negative metastatic breast cancer with germline BRCA mutation [20,34]. Most recently, on 25 February 2020, the FDA approved Niraparib for treating patients with advanced or metastatic HER2-positive breast cancer who have received two or more prior anti-HER2-based regimens in the metastatic setting.

### 3.1. Sensitivity of PARPi in TNBC Cells with Germline BRCA Mutations 

Since TNBCs have limited treatment options, we first tested the potential benefit of PARPi for TNBC cells. First, we examined the efficacy of the different PARPi in TNBC cells with germline BRCA mutations. In this study, we tested two TNBC cell lines with germline BRCA mutations, MDA-MB-436 and HCC1937. The MDA-MB-436 cell line was established from a metastatic breast cancer patient with a BRCA1 mutation at nt5396 leading to a truncated protein, and another cell line was HCC1937, which was derived from a patient with TNM (Tumor, Node, Metastasis) stage IIB primary cancer with the BRCA1 mutation at nt5832insC resulting in a frameshift change (ATCC^®^ TCP-1003).

Among the FDA-approved four PARPi, MDA-MB-436 showed the most response to Talazoparib (IC_50_ ≈ 0.13 µM) (Figure 1A,B). The cells were also sensitive to Rucaparib, Niraparib, and Olaparib with IC_50_ at 2.3, 3.2, and 4.7 µM (Figure 1A,B). Also, MDA-MB-436 was responsive to AZD2461, which targets PARP 1, 2, 3, at 1.7 µM (Figure 1B). However, the inhibitory activity of MDA-MB-436 to Talazoparib, Niraparib, and Rucaparib in HCC1937 was 10, 11, and 13 µM, respectively (Figure 1B). Data from this study showed that HCC1937 was less sensitive to Olaparib (IC_50_ ≈ 96 µM). Among the tested PARPi, HCC1937 was most responsive to PJ34HCl (IC50 ≈ 4 µM) (Figure 1B). The data suggests that even for the breast cancer cells having germline BRCA mutations, the sensitivity to the same PARPi could differ.

### 3.2. Sensitivity of PARPi in TNBC Cells without Germline BRCA Mutations

Next, we evaluated the efficacy of those PARPi in metastatic TNBC cells without germline BRCA mutations. Our data showed that MDA-MB-231 and MDA-MB-468 cancer cell lines, established from metastatic breast cancers, are susceptible to Talazoparib inhibition. The IC_50_ values were around 0.48 and 0.8 µM, respectively (Figure 2). Both MDA-MB-231 and MDA-MB-468 were responsive to Niraparib, Olaparib, and Rucaparib at ≤20 and <10 µM, correspondingly (Figure 2). Besides these four inhibitors, MDA-MB-468 was very sensitive to UPF 1069 (IC_50_ ≈ 0.8 µM) and PJ34HCl (IC_50_ ≈ 1.3 µM) (Figure 2B).

The efficacy of the 13 PARPi was also tested on non-metastatic TNBC cells without germline BRCA mutations, BT459, HCC1143, HCC70, and HCC1806. Those cell lines were also responsive to several PARPi. Talazoparib was able to effectively inhibit cell viability of BT549 and HCC70 at 0.3 and 0.8 µM and of HCC1143 and HCC1806 at 9 and 8 µM (Figure 3A,B). BT549, HCC1143, and HCC70 were also sensitive to Niraparib at 7, 9, and 4 µM, respectively (Figure 3). Furthermore, HCC1806 cells were very sensitive to Rucaparib (IC_50_ ≈ 0.9 µM) and Olaparib (IC_50_ ≈ 1.2 µM). PJ34HCl was also revealed as an effective PARPi for those metastatic and non-metastatic TNBC cells without germline BRCA mutations (Figure 3). The data implies that the inhibitory activity of PARPi may not only be limited to breast cancer with germline BRCA mutations, but PARPi may also benefit metastatic TNBC without BRCA mutations. Further mechanism-based studies are warranted.

### 3.3. Sensitivity of PARPi in ER-Negative and HER2-Positive Breast Cancer Cells

In this study, we examined the two ER- and HER2+ breast cancer cell lines and their response to different PARPi. SKBR3, a breast cancer cell line, was established from a patient with metastatic breast cancer and HER2 overexpression (ATCC^®^ 30-4500K). Similarly, the JIMT1 breast cancer cell line was established from a HER2+ breast cancer patient who was known to be clinically resistant to trastuzumab treatment [36]. Both cell lines do not express ER and have no known BRCA-mutations. We observed from this study that both SKBR3 and JIMT1 were highly sensitive to Talazoparib with IC_50_ values of about 0.04 and 0.002 µM, respectively (Figure 4A,B).

Furthermore, Niraparib was able to inhibit 50% of cell growth in SKBR3 at 7.3 µM and JIMT1 at 10 µM (Figure 4A,B). These results are encouraging, suggesting HER2+ metastatic breast cancer cells resistant to trastuzumab could also benefit from current FDA-approved PARPi. Additional investigations and mechanism studies are currently ongoing in our laboratory.

### 3.4. Sensitivity of PARPi in ER-Positive Breast Cancer Cells

We examined the efficacy of PARPi in the ER+/HER2- breast cancer cell line, MCF-7, and an ER+/HER2+ cell line, BT474. Our data showed that MCF-7 was responsive to Talazoparib and Niraparib at 1.1 to 5.4 µM, and Rucaparib and Olaparib at about 10 to 11 µM (Figure 4C,D). MCF-7 was also sensitive to AZD2461 (IC_50_ ≈ 5.2 µM). As the data shows in Figure 4D, BT474 was most responsive to PJ34HCl (IC_50_ ≈ 1.5 µM), followed by Niraparib (IC_50_ ≈ 13 µM). In addition, BT474 was also relatively sensitive to AG-14361 (IC_50_ ≈ 14.4 µM) (Figure 4D).

Overall, the ER+/HER- MCF-7 cells were responsive to all four FDA-approved PARPi. The ER+/HER2+ BT474 cells were more likely to be sensitive to Niraparib.

Table 3 summarizes the estimated IC_50_ values of the tested 13 PARPi for the 12 breast cancer cell lines. The data presented in Table 3 shows that Talazoparib and Niraparib are effective PARPi (IC_50_ < 20 µM) for most of the cell lines tested, except for one or two cell lines. Furthermore, PJ34HCl was also an effective PARPi for the 12 breast cancer cell lines.

## 4. Discussion

PARP inhibitors are an emerging class of small-molecule anticancer agents that have shown efficacy against BRCA-mutated gynecological cancers such as an ovarian, fallopian tube, and primary peritoneal cancer [37]. The FDA approved the first PARP inhibitor in 2014 for ovarian cancer [38]. It was not until 2018 that two additional PARP inhibitors, Olaparib and Talazoparib, were approved as monotherapies for BRCA-mutated HER2-negative metastatic breast cancer [39,40]. Nonetheless, the currently supported PARPi for clinic use is still under clinical trials, and many of the newly developed PARPi are not yet used for clinical treatment.

Our study demonstrated differential inhibitory activities of PARPi tested on the various breast cancer cells. The inhibitory activity of PARPi showed growth inhibition, independent of BRCA status, suggesting the possibility that cell lines sensitive to PARP inhibitors might have the defects of other homologous recombination factors.

We have demonstrated in this study that metastatic TNBC cells, MDA-MB-436 (BRCA1- deficient), MDA-MB231, and MDA-MB-468, were all sensitive to Talazoparib, a PARPi that has been approved to treat HER2- breast cancer with BRCA-mutations. Furthermore, HER2+ breast cancer cells SKBR3 and JIMT were highly responsive to Talazoparib (IC_50_ at 0.04 µM for SKBR3 and 0.002 µM for JIMT1). The data suggest that the benefit of Talazoparib extends beyond BRCA-mutant breast cancer, toward metastatic TNBC or HER2-positive breast cancer without BRCA-mutations.

More and more studies have shown that PARP inhibition could also be beneficial for cancer cells with dysfunction of genes involved in the DNA damage response. Recently, genome-wide profiling of synthetic genetic lethality identified CDK12 (Cyclin-dependent kinase 12) to be one of the additional genes conferring sensitivity to PARPi [41]. The study reported that the loss of CDK12 in ovarian cancer cells increased sensitivity to Olaparib treatment. Furthermore, the downregulation of CDK12 in HER2+ breast cancer cells also showed sensitivity to PARPi [42]. CDK12 plays an essential role in DNA repair and the maintenance of genomic stability [43,44]. CDK12 mutations were identified in 1.5% of TNBC patients [45,46], and the gene disruption was found in 13% of HER2+ breast cancer [6,47]. Furthermore, SLFN11 (Schlafen Family Member 11) is sensitive to PARPi in small cell lung cancer [48].

In contrast, our data showed that BRCA-mutant TNBC HCC1937 has less response to PARPi compared to BRCA-mutant TNBC MDA-MB-436 and other lines. The status of BRCA is not the only biomarker of response to PARPi. It is crucial to explore predictive biomarkers beyond BRCA1/2 for assessing sensitivity to PARPi.

Interestingly, trastuzumab-resistant cells such as JIMT1 displayed a high sensitivity to Talazoparib (at 0.002 µM) in our study. This finding is significant since trastuzumab is commonly used for treating HER2+ breast cancer patients, but many of these patients will eventually develop resistance to trastuzumab [7]. The data from our study suggests that inhibition of PARP could confer sensitivity to trastuzumab. Further studies are warranted.

It is well known that African American women are more likely to suffer from TNBC with poor disease outcome [3,4]. However, many of them may not be BRCA-mutation carriers; therefore, those patients will not be candidates for PARPi treatment clinically, at present. In this study, we tested the inhibitory potency of the 13 PARPi in two TNBC cell lines, HCC70 and HCC1806, generated from African American women with breast cancer and with wild-type BRCA (ATCC.org). The data from our study showed that HCC70 was very sensitive to both Talazoparib and G007-LK (both at 0.8 µM), and HCC1806 was most responsive to AZD2461 and Rucaparib (≤0.9 µM). Further studies that characterize the genomic signatures specific to breast cancer cell lines and tumors from African American patients may identify unique predictive biomarkers that more accurately represent the response to PARPi in African American women with TNBC.

PARP trapping activity may explain differences in the antitumor activity of each PARPi. PARP trapping is the ability of PARP to dissociate from damaged DNA to facilitate repair. Thus, PARP inhibition interferes with PARP dissociation and also inhibits PARP’s catalytic activity. Among the PARPi, PARP trapping potency varies widely. Talazoparib exhibits the highest PARP trapping ability out of the PARPi studied, followed by niraparib, olaparib, rucaparib, and veliparib.

Interestingly, trapping potency was not correlated with PARP catalytic inhibition [49,50,51,52]. Table 1 also reiterates that the PARPi have different PARP targets. Further studies are needed to analyze the selectivity of PARPi, pharmacokinetics, toxicity, and metabolism.

Multiple factors beyond the presence of a BRCA mutation may confer sensitivity to PARPi, including BRCAness (defecting in homologous recombination repair, mimicking BRCA1 or BRCA2 loss) and HRD. However, many clinical trials recruit patients based on their BRCA mutation status, subtype, or stage of the disease, but do not incorporate HRD testing in their studies. Currently, BRCA is the most widely used biomarker to assess sensitivity to PARPi. Cells that show a response to PARPi may also have a defect in homologous recombination repair genes, thus contributing to the HRD score. Instead of quantifying the effect of each genetic variation in the HR pathway, researchers have developed methods to score the competency of the HR pathway. Three scoring systems have emerged: HRD-loss of heterozygosity (HRD-LOH), homologous recombination defect large-scale transition (HRD-LST), and HRD-telomeric allelic imbalance (HRD-TAI) [53,54]. Another method called HRDetect utilizes whole-genome sequencing to predict BRCA deficiency based on mutational signatures [55]. A recent study evaluated an algorithm which generates a gene expression signature to predict PARPi response in cancer cell lines, patient-derived tumor cells, and patient-derived xenografts [56]. Through the integration of novel HRD biomarkers and scoring systems, identification of patient populations who may have therapeutic sensitivity to PARPi may be an advantage.

However, in consideration of all factors discussed previously, PARP proteins have other functions beyond DNA repair, such as chromatin remodeling, replication, and recombination [57]. To further elucidate more comprehensive clinical applications of PARPi as a therapeutic modality, extensive mechanistic studies are warranted.

The data from our current study serves as a basis for examining the effectiveness of different PARP inhibitors for multiple breast cancer subtypes. Further research into additional PARP targets is warranted to better understand the mechanisms behind PARP inhibition specific to either BRCA-positive or negative tumors. The data from this study indicate that PARP inhibition may be a useful therapeutic strategy for the treatment of BRCA-mutation-associated tumors as well as for a broader range of tumors that display characteristics of BRCAness or have dysfunctional genes involved in the DNA damage response.

## 5. Conclusions

Overall, our data showed that all four FDA-approved PARPi, Talazoparib, Niraparib, Olaparib, and Rucaparib displayed high inhibitory potency on different subtypes of breast cancer cells. In addition, PJ34HCl has shown good inhibitory strength in the inhibition of cell viability in different subtypes of breast cancer cells in this study as well.

## Figures and Tables

**Figure 1 jcm-09-00940-f001:**
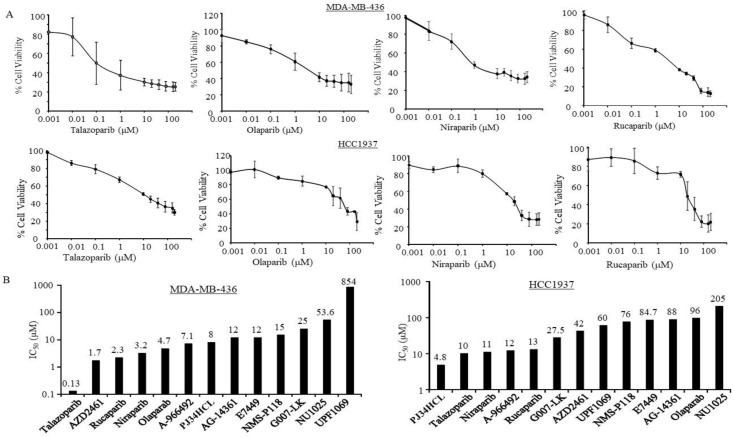
BRCA1 (BRast CAncer 1 gene) mutant TNBC (triple negative breast cancer) cells in response to Poly (ADP-ribose) polymerase inhibitors (PARPi). MDA-MB-436 and HCC1937 cells were treated with PARPi from 0.001 to 200 µM for 7 days, then the cell viabilities and the half maximal inhibitory concentration (IC_50_) of the indicated PARPi were determined as described in the Methods Section. (**A**) Inhibition curves in the cells under the indicated PARPi treatments, the points are averaged from 3 independent assays; bars represent standerd errors (SE). (**B**) The bars indicate mean IC_50_ of the indicated PARPi estimated by using the semi logarithm-transformed dose-response data with a Linear Regression model. The number presented on top of the bars indicates the mean IC_50_ for the indicated PARPi.

**Figure 2 jcm-09-00940-f002:**
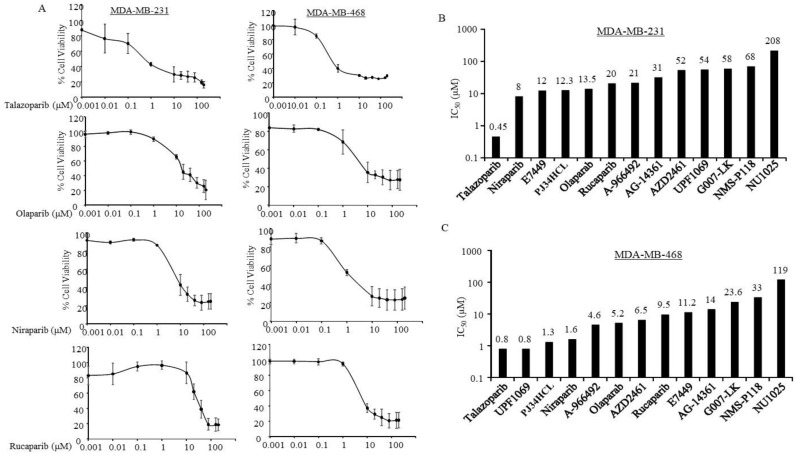
Metastatic triple negative breast cancer cells without BRCA1 (BReast CAncer 1 gene) mutation in response to PARPi. MDA-MB-231 and MDA-MB-468 were treated with Poly (ADP-ribose) polymerase inhibitors (PARPi) from 0.001 to 200 µM for 7 days, and then the cell viabilities and IC_50_ of the indicated PARPi were determined as described in the Methods Section. (**A**) Inhibition curves in the cells under the indicated PARPi treatments. Points: mean of three and bars represent standard errors. (**B**,**C**) The bars indicate mean of the half maximal inhibitory concentration (IC_50_) of the indicated PARPi for MDA-MB-231 (B) and MDA-MB-468 (C) cells. The IC_50_ values were estimated by using the semi logarithm-transformed dose-response data with a Linear Regression model. The number presented on top of the bars indicate the mean IC_50_ for the indicated PARPi.

**Figure 3 jcm-09-00940-f003:**
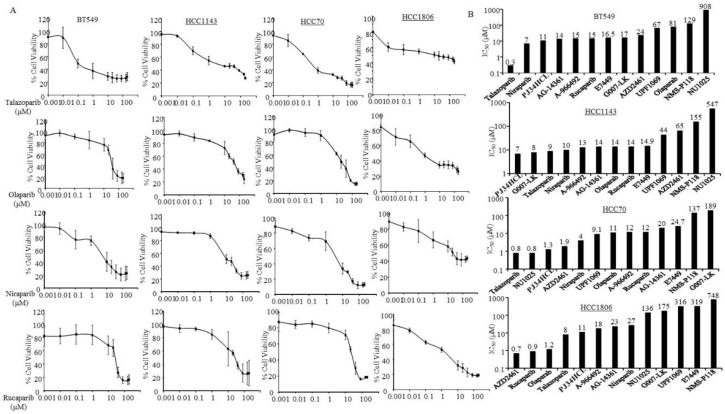
Triple negative non-metastatic breast cells without BRCA1 (BReast CAncer 1 gene) mutation in response to PARPi. BT549, HCC1143, HCC70, and HCC1806 were treated with Poly (ADP-ribose) polymerase inhibitors (PARPi) from 0.001 to 200 µM for 7 days, and then the cell viabilities and the half maximal inhibitory concentration (IC_50_) of the indicated PARPi were determined as described in the Methods Section. (**A**) Inhibition curves in the cells under the indicated PARPi treatments. Points: mean of three and bars represent standard errors. (**B**) The bars indicate mean IC_50_ of the indicated PARPi estimated by using the semi logarithm-transformed dose-response data with a Linear Regression model. The number presented on top of the bars indicates the mean IC_50_ for the indicated PARPi.

**Figure 4 jcm-09-00940-f004:**
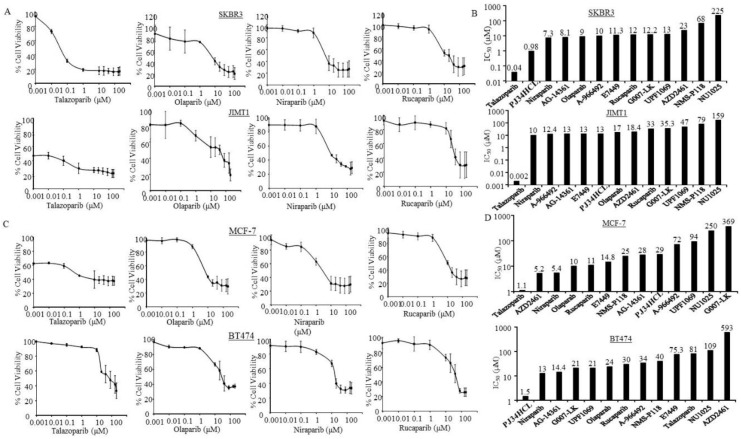
Breast cancer cells with different estrogen receptor (ER) and human epidermal growth factor receptor 2 (HER2) in response to Poly (ADP-ribose) polymerase inhibitors (PARPi). SKBR3, JIMT1, MCF-7, and BT474 cells were treated with PARPi from 0.001 to 200 µM for 7 days, and then the cell viabilities and the half maximal inhibitory concentration (IC_50_) of the indicated PARPi were determined as described in the Methods Section. (**A**) Inhibition curves of SKBR3 and JIMT1 cells under the indicated PARPi treatments. Points: mean of three and bars represent standard errors (SE). (**B**) The bars indicate mean IC_50_ of the indicated PARPi estimated by using the semi logarithm-transformed dose-response data with a Linear Regression model. The number presented on top of the bars shows the mean IC_50_ for the indicated PARPi. (**C**) Inhibition curves of MCF-7 and BT474 cells under the indicated PARPi treatments. Points: mean of three and bars represent SE. (**D**) The bar graphs of SKBR3 and JIMT1 cells show mean IC_50_ of each PARPi, calculated as described in the Methods Section.

**Table 1 jcm-09-00940-t001:** Targets of PARP * inhibitors used in this study.

PARP Inhibitors	PARP1	PARP2	PARP3	PARP5a/TNKS1 ^	PARP5b/TNKS2 ^§^
A-966492 [27]	++++	+++			
AG-14361 [27]	++				
AZD2461 [27]	++	+++		+	+
E7449 [27]	++++	++++			
G007-LK [27]				+	+
Niraparib [27]	+++	+++			
NMS-P118 [27]	++				
NU1025 [28]	+				
Olaparib [27]	++	++++			
PJ34-HCl [29]	++	++			
Rucaparib [30]	+++	+++	+++		
Talazoparib [27]	++++				
UPF-1069 [27]	+	++			

* Poly (ADP-ribose) polymerase; ^ Tankyrase 1; ^§^ Tankyrase 2. There are 18 members of the PARP family [31]. PARP1, PARP2, PARP5a/TNKS1, and PARP5b/TNKS2 are poly-ADP-ribosyl transferases, while PARP3 is a mono-ADP-ribosyl transferase [32]. These PARPs are involved in PARP enzymatic activity [31,33]. “+” indicates an inhibitory effect. A higher “+” designation marks increased inhibition.

**Table 2 jcm-09-00940-t002:** Characteristics of cell lines used in this study.

Cell Line	Subtype	BRCA Mutations	Derivation	Disease
MDA-MB-231	TNBC-Mesenchymal-like	No	51 years Caucasian female	Adenocarcinoma
MDA-MB-436	TNBC- Mesenchymal-like	Yes; 5396 + 1 g > A/truncated protein	43 years Caucasian female	Adenocarcinoma
MDA-MB-468	TNBC basal-like	No	51 years Black female	Adenocarcinoma
MCF-7	Luminal A	No	69 years Caucasian female	Adenocarcinoma
HCC1143	TNBC basal-like-1	No	52 years Caucasian female	TNM stage IIA, grade 3, primary ductal carcinoma
HCC1937	TNBC basal-like-1	Yes; 5382 insC/frameshift	23 years Caucasian female	TNM stage IIB, grade 3, primary ductal carcinoma
BT474	Luminal B	No	60 years Caucasian female	Ductal carcinoma
BT549	TNBC-Mesenchymal	No	72 years Caucasian female	Ductal carcinoma
SKBR3	HER2+ (sensitive to Herceptin)	No	43 years Caucasian female	Ductal carcinoma
JIMT1	HER2+ (resistant to trastuzumab)	No	62 years Caucasian female	Adenocarcinoma
HCC70	TNBC basal-like	No	49 years Black female	TNM stage IIIA, grade 3, primary ductal carcinoma
HCC1806	TNBC basal-like	No	60 years Black female	TNM stage IIB, grade 2, primary acantholytic squamous cell carcinoma

The cell lines represent various breast cancer subtypes, BRCA mutation status, patients of different ages and ethnicities, and various disease stages [35].

**Table 3 jcm-09-00940-t003:** ^ IC_50_ values of the tested * PARP inhibitors (PARPi) for each of the 12 cell lines.

Estimated IC_50_ (µM)												
Cell Lines	MDA-MB436	HCC1937	MDA-MB231	MDA-MB468	BT549	HCC1143	HCC70	HCC1806	SKBR3	JIMT1	MCF7	BT474
PARPi												
A-966492	7.1	12	21	4.6	15	13	12	18	10	12.4	72	34
AG-14361	12	88	31	14	14	14	20	23	8.1	13	28	14.4
AZD2461	1.7	42	52	6.5	24	65	1.9	0.7	23	18.4	5.2	593
E7449	12	84.7	12	11.2	16.5	14.9	24.7	319	11.3	13	14.8	75.3
G007-LK	25	27.5	58	23.6	17	8	189	175	12.2	35.3	369	21
Niraparib	3.2	11	8	1.6	7	10	4	27	7.3	10	5.4	13
NMS-P118	15	76	68	33	129	155	137	748	68	79	25	40
NU1025	53.6	205	208	119	908	547	0.8	136	225	159	250	109
Olaparib	4.7	96	13.5	5.2	81	14	11	1.2	9	17	10	24
PJ34HCL	8	4.8	12.3	1.3	11	7	1.3	11	0.98	13	29	1.5
Rucaparib	2.3	13	20	9.5	15	14	12	0.9	12	33	11	30
Talazoparib	0.13	10	0.45	0.8	0.3	9	0.8	8	0.04	0.002	1.1	81
UPF1069	854	60	54	0.8	67	44	9.1	316	13	47	94	21

^ the half maximal inhibitory concentration, * Poly (ADP-ribose) polymerase inhibitors.

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
