# Peer review of "Response of Breast Cancer Cells to PARP Inhibitors Is Independent of BRCA Status"

_jcm, 2020, doi:10.3390/jcm9040940_

Round 1
Reviewer 1 Report
Since it has been reported that inhibitors of DNA repair enzyme, poly (ADP-ribose) polymerase (PARP), cause profound cell inhibitory effects on BRCA1 or BRCA2-deficient cells, a number of PARP inhibitors have been developed. Then, several PARP inhibitors have been approved recently and used to treat breast cancer and ovarian cancer. However, it has been shown that a certain percentage of patients with wild-type BRCA1/2 can still get benefit from PARP inhibitors.
The authors analyzed the inhibitory effect of 13 different PARP inhibitors on cell growth of 12 breast cancer cell lines with and without BRCA mutation. They found that triple negative breast cancer (TNBC) cell lines were very sensitive to PARP inhibitors regardless of the BRCA-status. Moreover, the ER-negative/HER2-positive (ER-/HER2+) cell lines were highly sensitive to Talazoparib.
This is interesting study and will contribute to improve the treatment with PARP inhibitors for TNBC and ER-/HER2+ breast cancers. The manuscript is well-written, the experiments are executed to a high standard, and to a large extent the conclusions drawn by the authors are supported by the results they present.
General Comments:
- The authors conclude that PARP inhibitors therapy will be effective to metastatic TNBCs and ER-/HER2+ breast cancers in addition to cancers with BRCA-mutations. The authors described about only mutations of BRCA. However, reduction of BRCA expression were also found in breast cancer. If the expression levels of BRCA1 and BRCA2 are associated with the sensitivity to PARP inhibitors, those results will be interesting.
- In the section of Discussion, the authors should mention clearly about the possibility that cell lines sensitive to PARP inhibitors might have the defects of other homologous recombination factors.
- It has been reported that mechanisms of action are different between PARP inhibitors. The authors should describe them in the section of Discussion.
Specific Comments:
- The authors had better present the table that show the characteristic of twelve cell lines used in this study.
- In Figure legend of Figure 3, ‘‘TNBC non-metastatic cells with BRCA1 mutation’’ should be ‘‘TNBC non-metastatic cells without BRCA1 mutation’’.
Author Response
Responses to Reviewer 1 comments:
Comments from Reviewer 1:
Since it has been reported that inhibitors of DNA repair enzyme, poly (ADP-ribose) polymerase (PARP), cause profound cell inhibitory effects on BRCA1 or BRCA2-deficient cells, several PARP inhibitors have been developed. Several PARP inhibitors have been approved recently and used to treat breast cancer and ovarian cancer. However, it has been shown that a certain percentage of patients with wild-type BRCA1/2 can still get benefit from PARP inhibitors.
The authors analyzed the inhibitory effect of 13 different PARP inhibitors on cell growth of 12 breast cancer cell lines with and without BRCA mutations. They found that triple-negative breast cancer (TNBC) cell lines were very sensitive to PARP inhibitors regardless of the BRCA-status. Moreover, the ER-negative/HER2-positive (ER-/HER2+) cell lines were highly sensitive to Talazoparib.
This is an interesting study and will contribute to improving the treatment with PARP inhibitors for TNBC and ER-/HER2+ breast cancers. The manuscript is well-written, the experiments are executed to a high standard, and to a large extent, the conclusions drawn by the authors are supported by the results they present.
Response: We are very grateful to the reviewer for encouraging comments. We have addressed all of the concerns in the revised manuscript and updated all references.
General Comments:
- The authors conclude that PARP inhibitors therapy will be effective in metastatic TNBCs and ER-/HER2+ breast cancers in addition to cancers with BRCA-mutations. The authors described about only mutations of BRCA. However, the reduction of BRCA expression was also found in breast cancer.
Response: In the revised manuscript, in the introduction section, we have included additional information and references on decreased BRCA1 expression in sporadic breast cancer. Methylation of BRCA promoter and sporadic loss of the wild-type allele could contribute to BRCA inactivation in breast cancer without BRCA mutations. Please see the introduction section in the revised manuscript in track changes (lines 68-78).
- If the expression levels of BRCA1 and BRCA2 are associated with the sensitivity to PARP inhibitors, those results will be interesting.
Response: We agree. Since we have a very short response time (7 days) we are not able to conduct at this time, additional experiments to directly look if the expression levels of BRCA1 and BRCA2 are associated with the sensitivity to PARP inhibitors in those tested cell lines. We will examine this association in our ongoing study.
A study from Elstrodt et al. (Cancer Res 2006, 66:41-5) conducted a BRCA1 mutation analysis of 41 human breast cancer cell lines and found BRCA1 allelic loss in most breast cancer cells with or without mutations. The allelic loss affected BRCA1 transcript expression, such as MCF-7 cells have no mutations of BRCA1 but showed BRCA1 allelic loss. The BRCA1 transcript expression was barely detectable in MCF-7 cells. This is consistent with the data from our study that MCF-7 cells were able to respond to PARPi treatment.
- In the section of Discussion, the authors should mention clearly about the effect of homologous recombination.
Response: We have included in the discussion section on the effect of homologous recombination deficiency. We have stated that cells which show a reaction to PARPi may also have a defect in homologous recombination repair genes, thus contributing to HRD score. These changes are reflected in lines: 264-265 and 311-329 in the Discussion of the revised manuscript (in track changes).
- It has been reported that mechanisms of action are different between PARP inhibitors. The authors should describe them in the section of Discussion.
Response: We agree. We have described the role of PARP trapping activity from the different PARPi tested in our study, in lines 303-310 in the Discussion section.
Specific comments:
- The authors had better present the table that show the characteristic of twelve cell lines used in this study.
Response: We have added a “Table 2. Characteristics of cell lines used in this study” in the methods section under “Cell lines and cell culture” in the revised manuscript.
- In Figure legend of Figure 3, “TNBC non-metastatic cells with BRCA1 mutation” should be “TNBC non-metastatic cells without BRCA1 mutation”.
Reviewer 2 Report
The current studies is describing the impact of PARP inhibitory molecules on different breast cancer cell lines. The study was well executed and is very important for the treatment of so far not easyly treatable types of breast cancer. I would recommand some minor changes which makes it reliable for readers which are not from the scientific field. Please check the comments made in the attached PDF.

Author Response
Responses to Reviewer 2 comments
- English language and style are fine/minor spell check required
Response: We have made an additional check on English grammar and spelling, and made the necessary corrections
- I think BRCA-mutations needs a deeper introduction as it is a major point in your chain of arguments.
Response: Yes, Thank you. We have made significant changes in the introduction by adding more information on BRCA-mutations and loss of heterozygosity of BRCA1 in somatic breast cancer (lines: 46-62, 70-79).
We also provide more detailed information for Table 1 and figures in figure legends in the revised manuscript, and all changes are tracked.
Round 2
Reviewer 1 Report
The authors adequately addressed my previous concerns and the revised manuscript improved. However, there are some points that should be corrected.
- In Line 43, ‘‘Herceptin’’ should be ‘‘trastuzumab’’.
- In Line 44, ‘‘PARP inhibitors’’ should be ‘‘PARP inhibitors (PARPi)’’.
- In Line 210, the paragraph that start with ‘‘The efficacy of the 13 PARPi…’’ in the Legend of Figure 2 should be in the main text.
- In Line 263, ‘‘Table 2’’ should be ‘‘Table 3’’
- The sentence of Lines 278-279 should be removed. In this manuscript, the authors described about the possibility that cell lines sensitive to PARP inhibitors might have the defects of other homologous recombination factors in Lines 333-334.
- In Table 2, ‘‘Herceptin’’ should be ‘‘trastuzumab’’.
Author Response
Responses to Reviewer 1 comments:
- In Line 43, ‘‘Herceptin’’ should be ‘‘trastuzumab’’.
Response: In previous line 43 (current Line 38) we deleted “(Herceptin)” and changed “trastuzumab (Herceptin)” to “trastuzumab”
- In Line 44, ‘‘PARP inhibitors’’ should be ‘‘PARP inhibitors (PARPi)’’.
Response: In previous line 44 (current Line 40), “(PARPi)” has been added after “PARP inhibitors”
- In Line 210, the paragraph that start with ‘‘The efficacy of the 13 PARPi…’’ in the Legend of Figure 2 should be in the main text.
Response: In previous line 210, the paragraph in the Legend of Figure 2 has been move to main text.
- In Line 263, ‘‘Table 2’’ should be ‘‘Table 3’’
Response: In previous line 263 (current line 253), “Table 2” has been corrected as “Table 3”
- The sentence of Lines 278-279 should be removed. In this manuscript, the authors described about the possibility that cell lines sensitive to PARP inhibitors might have the defects of other homologous recombination factors in Lines 333-334.
Response: The sentence of previous lines 278-279 has been removed and references numbers cited have been corrected
- In Table 2, ‘‘Herceptin’’ should be ‘‘trastuzumab’’.
Response: In Table 2, “Herceptin” has been changed to “trastuzumab”